# Achieving Personalized Privacy-Preserving Graph Neural Network via Topology Awareness

## Abstract

Graph neural networks (GNNs) with differential privacy (DP) offer a reliable solution for safeguarding sensitive information within graph data. Nonetheless, existing DP-based privacy-preserving GNN learning frameworks generally overlook the local topological heterogeneity of graph nodes and tailor the same privacy budget for all nodes, which may lead to either overprotection or underprotection of some nodes, potentially diminishing model utility or posing privacy leakage risks. To address this issue, we propose a **T**opology-aware **D**ifferential **P**rivacy **G**raph **N**eural **N**etwork learning framework, termed **TDP-GNN**, which can achieve personalized privacy protection for each node with improved privacy-utility guarantees. Specifically, **TDP-GNN** first identifies the topological importance of each node via an adjacency information entropy method. Then, the personalized topology-aware privacy budget is designed to quantify the privacy sensitivity of each node and adaptively allocate the privacy protection strength. Besides, a weighted neighborhood aggregation mechanism is proposed during the message-passing process of GNN training, which can eliminate the impact of the introduced differentiated DP noise on the utility of the GNN model. Since **TDP-GNN** is based on node-level local DP, it can be seamlessly integrated into any GNN architecture in a plug-and-play manner while ensuring formal privacy guarantees. Theoretical analysis indicates that **TDP-GNN** achieves $\epsilon$-differential privacy over the entire graph nodes while providing personalized privacy protection. Extensive experiments demonstrate that **TDP-GNN** consistently yields better utilities when applied to various GNN architectures (e.g., GCN and GraphSAGE) across a diverse set of benchmarks.

## CCS Concepts

• **Security and privacy** → **Privacy protections**; **Privacy-preserving protocols**; **Social network security and privacy**.

## Keywords

Graph Neural Networks; Privacy-Preserving; Differential Privacy; Topology Awareness;

**ACM Reference Format:**
Anonymous Author(s). 2024. Achieving Personalized Privacy-Preserving Graph Neural Network via Topology Awareness. In *Proceedings of the ACM*

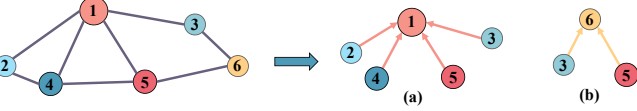

Global graph topology       Local computational subgraphs

**Figure 1: The illustration of topological heterogeneity of each node in a graph. As shown in (a) and (b), due to the heterogeneity in topological structures of nodes, their local computational subgraphs during the message-passing process of GNN training are different: (a) Node 1 needs to aggregate feature data from four neighboring nodes {2, 3, 4, 5}, (b) while node 6 can only need aggregate feature data from two neighboring nodes {3, 5}.**

*Web Conference 2024 (Conference acronym 'XX)*. ACM, New York, NY, USA, 9 pages. https://doi.org/XXXXXXX.XXXXXXX

## 1 Introduction

Graph Neural Networks (GNNs) have shown superior performance in learning graph representations and have been applied in a variety of fields, such as intrusion detection, drug discovery, and social recommendation systems [1–5]. However, most real-world graphs linked to human activities, such as economic and social networks [6, 7], often contain personal data and may involve sensitive information. For instance, a user's profile information, favorites, friend lists, likes, and comments on social networks may be private to the user. Direct learning from raw graph data may lead to the exposure of private sensitive information, which may conflict with the data protection legislation such as the General Data Protection Regulation (GDPR) [8].

To mitigate potential privacy leakage [9, 10], a feasible approach is to employ differential privacy (DP) technology [11]. By injecting noise into sensitive information (e.g., node features and labels) , DP can effectively safeguard user privacy against unauthorized exposure. Recently, several DP-based privacy-preserving GNN learning methods [12–18] have been proposed. For example, Sajadmanes et al. [12] introduced a locally private GNN model in a distributed learning framework, ensuring the confidentiality of node features and labels. However, this framework does not apply to scenarios where the graph edges must remain private. To protect the edge privacy, Wu et al. [14] proposed an edge-level differentially private GNN learning algorithm by perturbing the graph's adjacency matrix. However, this method exhibits suboptimal performance and is frequently outperformed by multi-layer perceptron models (MLPs) that are trained entirely without link information [19]. Furthermore, Pei et al. [18] inject local differential privacy noise into decentralized local graphs to protect both node feature privacy and edge privacy in GNN learning.

Nonetheless, existing methods overlooked the topological heterogeneity of different nodes and assumed that all nodes have the same privacy sensitivities, i.e., setting a uniform privacy budget for all nodes. This "one-size-fits-all" approach is not suitable for real-world complex networks, as graph nodes in real-world networks typically display intricate topological structures with diverse privacy sensitivity. As shown in Figure 1, there are topology heterogeneities in the local computational structure of different nodes, which lead to the neighborhood information aggregated by GNN in the message-passing process being different. Overlooking the heterogeneity of node topological structures in privacy-preserving graph learning may result in over-protection or under-protection of certain nodes. This is because nodes with higher degrees typically exhibit stronger network influence and usually require a higher level of privacy protection. Using a uniform privacy budget may not satisfy their privacy requirements. In addition, nodes with fewer neighbors are more susceptible to DP noise from their neighboring nodes. As a result, under the same privacy protection strength, the data utility of lower-degree nodes is more easily compromised. In light of this, it is necessary to design a topology-aware differential privacy GNN learning framework to provide personalized privacy protection for each node.

**Challenges.** Achieving effective GNN learning from complex graph data while providing topology-aware personalized privacy protection for each node is inherently challenging. On the one hand, the topological importance of each node varies and is difficult to measure with a specific value, hence accurately assessing the required strength of privacy protection for each node is challenging. On the other hand, the unique message-passing mechanism in GNN requires each node to aggregate neighboring information to update its representation, while the personalized perturbations of neighboring features may introduce different degrees of DP noise. Therefore, it is also challenging to obtain a high-accuracy GNN model trained from differentiated noisy graph data.

To address the above challenges, we present a topology-aware differential privacy graph neural network learning framework named TDP-GNN, which can achieve personalized privacy protection based on the topological properties of each node while maintaining high model utility. Specifically, we first introduce a *node importance identification mechanism* based on Adjacency Information Entropy (AIE) [20]. This mechanism calculates the topological importance of nodes based on the associations between the node and its direct and indirect neighboring nodes. Then, a *topology-aware privacy budget allocation mechanism* is designed to quantify the privacy sensitivity of each node according to its topological importance and adaptively allocate a specific privacy budget to each node to provide corresponding privacy protection strength. Furthermore, to improve the accuracy of the GNN model, a *weighted neighborhood aggregation mechanism* is proposed, which can suppress noisy data interference by assigning adaptive weights to each neighboring node so as to obtain an effective representation of the central node.

The specific contributions of this work are as follows:

- We present a topology-aware differential privacy graph neural network learning framework named TDP-GNN. To the best of our knowledge, this is the first work that provides personalized privacy protection for nodes based on their topological characteristics in graph learning.
- We measure the importance of nodes with the consideration of the local topological structure and allocate personalized privacy budgets for each node. Additionally, we design a weighted neighborhood aggregation mechanism during the message-passing process in GNN, which can enhance the accuracy of GNN models trained from noisy data with different levels of perturbations.
- We provide a theoretical analysis demonstrating that TDP-GNN can achieve $\epsilon$-differential privacy while offering personalized privacy protection for each node. Additionally, extensive experiments show that TDP-GNN consistently yields performance improvement when applied to various GNN architectures across a diverse set of benchmarks. For example, experiments show that plugging in TDP-GNN to GCN and GraphSAGE improves by an average of 4.90% and 5.37% in terms of accuracy on Cora, Citeseer, Pubmed, and Facebook.

## 2 BACKGROUND

### 2.1 Graph Neural Network

Let $\mathbb{G} = \{\mathcal{N}, \mathcal{E}, \mathbf{A}, \mathbf{X}, \mathbf{Y}\}$ be an undirected and unweighted graph dataset consisting of node set $\mathcal{N}$ and edge set $\mathcal{E}$ represented by a binary adjacency matrix $\mathbf{A} \in \{0,1\}^{N \times N}$, where $N = |\mathcal{N}|$ denotes the number of nodes, and $\mathbf{A}_{u,v} = 1$ if there is an edge $(u,v) \in \mathcal{E}$ between node $u$ and node $v$. The feature $\mathbf{X}_u \in \mathbf{R}^k$ of node $u \in \mathcal{N}$ is a $k$-dimension vector. $\mathbf{Y} \in \{0,1\}^{N \times C}$ represents the label of nodes, and $C$ is the class number.

A typical GNN with $d$ layers consists of $d$ graph convolution layers arranged sequentially. The representation for a node $u$ at the $i$-th layer initially involves aggregating the representations of its neighboring nodes at the $(i-1)$-th layer. Subsequently, it is followed by a learnable transformation, as outlined below:

$$\Phi_u^{(i)} = Update(Aggregate(\{\Phi_v^{(i-1)} : \forall v \in \mathcal{N}_u\}), \Theta^{(i)}) \quad (1)$$

where $\mathcal{N}_u = \{v : \mathbf{A}_{u,v} \neq 0\}$ denotes the neighboring set of node $u$, and $\Phi_v^{(i-1)}$ denotes the representation of node $v$ at the $i-1$ layer. $Aggregate(\cdot)$ is a (sub)differentiable, permutation-invariant aggregator, which can be operations like MAX, MEAN, or SUM. Additionally, $Update(\cdot)$ is a learnable function, such as MLP [21], parameterized by $\Theta^{(i)}$ that processes the aggregated vector to produce the new representation $\Phi_n^{(i)}$.

Initially, we have $\Phi^{(0)} = \mathbf{X}$ (i.e., node features) as the input to the first layer of the GNN. The final layer generates an output representation vector for each node, which can be utilized in various downstream tasks [22–24].

### 2.2 Differential Privacy

Differential privacy has proven to be a strong and rigorous privacy framework for protecting privacy in data analysis across various applications. The formal definitions of differential privacy are stated as follows:

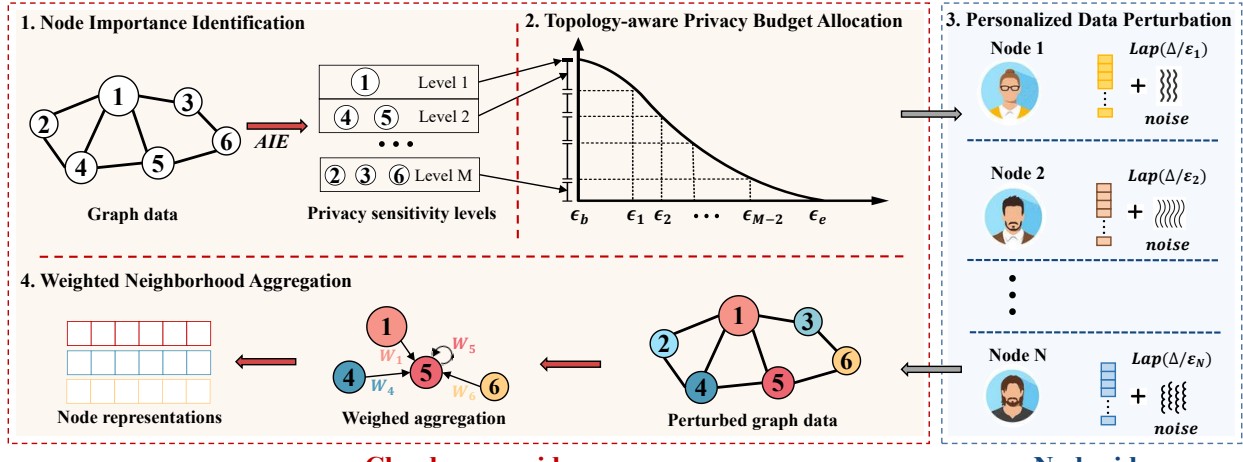

**Figure 2: The workflow of TDP-GNN: a topology-aware differential privacy GNN learning framework.**

**Definition 1.** *(Differential Privacy [25]).*

*A randomized algorithm $\mathcal{B}$ that takes as input a dataset consisting of individuals is $(\epsilon, \xi)$-differential private (DP) if for any pair of neighboring data $a, b$ that differ on one single record, and any Sets $S \in Range(\mathcal{B})$,*

$$Pr[\mathcal{B}(a) \in S] \le e^{\epsilon} \cdot Pr[\mathcal{B}(b) \in S] + \xi \qquad (2)$$

*and if $\xi = 0$, we say that $\mathcal{B}$ is $\epsilon$-differential private.*

The Laplace mechanism is a widely employed noise addition mechanism in the field of differential privacy.

**Definition 2.** *(Laplace Mechanism [26]). Let the sensitivity of the fuction $f : Q \to S$ be $\triangle(f) = \max |f(Q_2) - f(Q_1)|$ for all $Q_1, Q_2 \in Q$ such that $|Q_2 - Q_1| \le 1$. The mechanism $\mathcal{B} : Q \to S$ defined as follows satisfies $\epsilon$-differential privacy:*

$$\mathcal{B}(Q_1) = f(Q_1) + Laplace(0, \triangle(f)/\epsilon) \qquad (3)$$

### 2.3 Threat Model

As shown in Figure 2, the TDP-GNN includes two primary entities: the cloud server and the node users. In the threat model, the cloud server is considered honest but curious, which is a common assumption in many prior works [27, 28]. It has access to the node's adjacency matrix **A**, which provides the graph topology information. However, the cloud server cannot directly access the feature matrix **X** and labels **Y**, which are distributed among nodes and remain private. The cloud server adheres to the TDP-GNN protocol, but it may be curious about nodes' sensitive feature data and attempt to infer private information from it. Similarly, node users are regarded as honest but curious. They will follow the agreed protocol but may try to compromise the data privacy of other node users. In this scenario, it is crucial to ensure the confidentiality of the features and labels of nodes, mitigating the risk of unintentional disclosure to external entities.

## 3 Proposed METHOD

### 3.1 Overview of TDP-GNN

The goal of this paper is to design a topology-aware differential privacy framework to provide personalized privacy protection for each node during the GNN training process. Figure 2 illustrates the TDP-GNN workflow, which includes the process of node importance identification, topology-aware privacy budget allocation, and weighted neighborhood aggregation at the cloud server side, as well as the personalized data perturbation process at each node side. The node importance identification mechanism aims to measure the node's importance based on its local topological structure and subsequently determines the node's privacy sensitivity according to its importance score. The higher the node's importance score, the higher the privacy sensitivity of the node. The topology-aware budget allocation mechanism aims to allocate a specific privacy budget to provide personalized privacy protection based on the node's privacy sensitivity. Finally, the weighted neighborhood aggregation mechanism adjusts weights for neighborhood aggregation based on the importance of neighboring nodes. The principle behind this mechanism is that as the importance of a neighboring node increases, its privacy sensitivity also rises, resulting in more DP noise being injected into its feature data. Therefore, it is necessary to assign a smaller weight to these nodes during the aggregation phase. In doing so, the utility of the original node features can be better preserved. The detailed implementations of TDP-GNN are shown in Algorithm 1.

### 3.2 Node Importance Identification

In order to comprehensively measure the topological importance of nodes in the graph, we aim to introduce a node importance identification method based on Adjacency Information Entropy (AIE) [20]. Compared with traditional identification approaches, such as merely using node degree as the evaluation metric, this AIE-based approach not only considers the association between a node and its direct neighboring nodes but also considers the complex

---

**Algorithm 1:** TDP-GNN

---

**Input:** Graph $\mathbb{G} = \{\mathcal{N}, \mathcal{E}, \mathbf{A}, \mathbf{X}, \mathbf{Y}\}$, privacy budget $\{\epsilon_b, \epsilon_e, \epsilon_l\}$,
   privacy protection sensitivity level $M$
**Output:** Node's predicted label
1: // *Node importance identification:*
2: **for** $i \in \{1, 2, \cdots, N\}$ **do**
3:    Calculate $AIE_i$ for node $i \leftarrow$ Eq. (4), (5), and (6);
4: **end for**
5: // *Topology-aware privacy budget allocation:*
6: Divide nodes into $M$ levels of privacy sensitivities according to
   the value of $AIE$;
7: **for** $i \in \{1, 2, \cdots, N\}$ **do**
8:    Allocate a personalized privacy budget $\varepsilon_i$ for node $i$ based
   on its privacy sensitivity;
9: **end for**
10: // *Personalized data perturbation:*
11: **for** each node $i, i \in \{1, 2, \cdots, N\}$ **do**
12:    $\hat{\mathbf{X}}_i = f(\mathbf{X}_i) + Lap(\triangle f / \varepsilon_i)$;
13: **end for**
14: Perturb the corresponding labels $Y_i$ based on *random response*
   *mechanism* under $\epsilon_l \leftarrow$ Eq. (12);
15: // *Weighted neighborhood aggregation:*
16: **for** $k \in \{1, 2, \cdots, K\}$ **do**
17:    Calculate the weight $W_{i,j}$ of each neighbor of node $i$ and
   performs weighted aggregation $\leftarrow$ Eq. (10) and (11);
18: **end for**
19: **return** predicted node label $\mathbf{Y}_i'$;

---

relationship between a node and its indirect neighboring nodes, thereby obtaining better node importance identification results.

Formally, let $\mathcal{N}$ be the node set of the graph $\mathbb{G}$. The AIE of a node $u \in \mathcal{N}$ can be measured as follows:

(i) The Adjacency Degree $AD_u$ of a node $u$ quantifies its influence on its neighboring nodes. We measure $AD_u$ using the following equation:

$$AD_u = \sum_{v \in \mathcal{N}_u} D_v. \tag{4}$$

where $D_v$ represents the degree of the node $v$ and $\mathcal{N}_u$ denotes the neighborhood set of node $u$.

(ii) The Probability Function $p_u$ of node $u$ defines the probability of selecting node $u$ among its neighbors. It is expressed as follows:

$$p_u = \frac{D_u}{AD_v}, v \in \mathcal{N}_u. \tag{5}$$

(iii) The Adjacency Information Entropy $AIE_u$ of node $u$ quantifies the topological importance of node $u$:

$$AIE_u = -\sum_{v \in \mathcal{N}_u} (p_u \log_2 p_u) p_v. \tag{6}$$

## 3.3 Topology-aware Privacy Budget Allocation

The existing works generally overlook the topological heterogeneity of nodes and allocate a uniform privacy budget to all nodes. However, due to the differences in privacy sensitivity of nodes in

the real-world graphs, this "one-size-fits-all" approach may lead to overprotection or underprotection of some nodes. To address the above issues, we design a topology-aware privacy budget allocation mechanism to provide personalized privacy protection for each node with DP. In real-world networks, important nodes typically exhibit higher influence and thus require higher privacy protection strength. In DP techniques, a smaller privacy budget implies a higher degree of noise disturbance and provides a higher privacy protection strength. Thus, given the local topological heterogeneity of nodes in the network, a node with greater importance should be allocated a smaller privacy budget.

As studied in [29], the node distribution in real-world graphs typically follows a power-law distribution, where the majority of nodes possess relatively small social influence (i.e., low privacy sensitivity), while a minority of nodes may have significant social influence (i.e., high privacy sensitivity). Hence, we assume that the node's privacy budget is constrained within the range $[\epsilon_b, \epsilon_e]$ and follows an exponential distribution within the interval $[\epsilon_b, \epsilon_e]$. Consequently, the node's privacy budget could be sampled within the exponential distribution interval corresponding to the distribution of their privacy sensitivity.

However, due to the large scale of real-world graphs and the diverse privacy sensitivity of each node, it is difficult to efficiently map the privacy sensitivity to a specific privacy budget value. Therefore, we design a hierarchy-based personalized privacy budget allocation strategy. This strategy first divides the nodes' privacy sensitivities into different levels based on their calculated node importance scores, and then establishes a mapping relationship between the privacy sensitivity levels and the privacy budget sub-intervals, thereby accelerating the privacy budget allocation process. Specifically, suppose that the node privacy sensitivity can be divided into $M$ levels. We allocate $\epsilon_b$ to nodes with the highest privacy sensitivity level, and then the total privacy budget is divided into $M - 1$ sub-intervals: $(\epsilon_b, \epsilon_1], (\epsilon_1, \epsilon_2], ..., (\epsilon_{M-3}, \epsilon_{M-2}], (\epsilon_{M-2}, \epsilon_e]$. Subsequently, the cloud server could sample a privacy budget for each node from the corresponding sub-interval to match its privacy sensitivity level.

Based on the hierarchy-based personalized privacy budget allocation strategy, the crucial task is finding suitable boundary points $\epsilon_1, \epsilon_2, \cdots, \epsilon_{M-2}$ to determine the length of each sub-interval. A practical approach is to set the length of a sub-interval according to the proportion of the number of nodes in the corresponding privacy sensitivity level. Let the proportions of the number of nodes in the $M$ privacy sensitivity levels be: $\{\beta_1, \cdots, \beta_{M-1}, 1 - \sum_{i=1}^{M-1} \beta_i\}$. Given an exponential distribution $f(y, \lambda) = \lambda e^{-\lambda y}$, the domain of $y$ can be partitioned according to these proportions $\beta_1, \beta_2, \ldots, \beta_{M-1}$. By doing so, we can establish a mapping relationship between $y$ and $\epsilon$.

In particular, assuming the domain of $y$ is divided into $M + 1$ intervals $[0, y_b), [y_b, y_1), \ldots, [y_{M-2}, y_e), [y_e, +\infty)$, when sampling a value from the random variable $y$ on the exponential distribution, the probability of it falling within the interval $[y_b, y_1), \ldots, [y_{M-2}, y_e)$ should correspond to $\beta_1, \beta_2, \ldots, \beta_{M-1}$. Let $F(y, \lambda) = 1 - e^{-\lambda y} \ (y \geq 0)$ be the cumulative distribution function. Subsequently, $y_e$ should be mapped to $\epsilon_b$, $y_i$ to $\epsilon_{M-1-i}$ for $i \in \{1, \ldots, M - 2\}$, and $y_b$ to $\epsilon_e$. In detail, the values of $y_1$ and $y_e$ can be obtained through the

following formula:

$$y_1 = E_{1-\beta_1 - F(y_b)}, y_e = E_{1-\sum_{i=1}^{M-2} \beta_i - F(y_b)} \quad (7)$$

where $E_{1-\beta_1 - F(y_b)}$ denotes a $1 - \beta_1 - F(y_b)$-quantile that satisfies $P(y > y_1)$, and $E_{1-\sum_{i=1}^{M-2} \beta_i - F(y_b)}$ represents a $1-\sum_{i=1}^{M-2} \beta_i - F(y_b)$-quantile that fulfills $P(y > y_e) = 1 - \sum_{i=1}^{M-2} \beta_i - F(y_b)$.

Considering $y_b$ as typically close to 0, we assume $F(y_b) = 0$. Thus, we define the formula as follows:

$$y_1 = E_{1-\beta_1}, y_e = E_{1-\sum_{i=1}^{M-2} \beta_i} \quad (8)$$

After obtaining the values of $y_1$ and $y_e$, along with establishing the mapping between $y$ and $\epsilon$, we can derive the relationship $\frac{(\epsilon_e - \epsilon_{M-2})}{(\epsilon_e - \epsilon_b)} = \frac{y_1}{y_e}$. Within a fixed privacy budget interval $[\epsilon_b, \epsilon_e]$, we can ascertain the value of $\epsilon_{M-2}$. Other boundary points can be obtained using the same procedure. Once the boundary points $\epsilon_1, \epsilon_2, \ldots, \epsilon_{M-2}$ are obtained, the overall privacy budget interval division can be determined. Subsequently, the cloud server samples a specific privacy budget $\varepsilon_i$ for each node $i$ ($i \in \{1, 2, \ldots, N\}$) from its corresponding privacy budget sub-interval.

## 3.4 Personalized Data Perturbation

To protect the node feature privacy, we inject DP noise to perturb the node feature data. The personalized DP Perturbation is achieved by using the allocated personalized privacy budget for each node. The perturbed feature $\mathbf{X}_u$ of node $u$ is outlined below:

$$\hat{\mathbf{X}}_u = f(\mathbf{X}_u) + Laplace(\triangle f / \varepsilon_u) \quad (9)$$

## 3.5 Weighted Neighborhood Aggregation

Since the data utility will be degraded after injecting DP noise, we propose a weighted neighborhood aggregation mechanism to mitigate the impact of DP noise on the accuracy of the GNN model. As described in the ?? section, the personalized privacy budget depends on the importance of the node. When the node importance is higher, the allocated privacy budget is smaller, that is, the more perturbation noise is injected. Therefore, to suppress the injected perturbation noise, a smaller aggregation weight should be assigned to nodes with a higher importance.

Based on the above principle, the aggregation weight of a neighbor $v$ for node $u$ in the neighborhood aggregation process is as follows:

$$W_{u,v} = 1 + \frac{1}{D_u} - \frac{AIE_v}{\sum_{i=1}^{Ner_u} AIE_{u,i} + AIE_u} \quad (10)$$

where $Ner_u$ denotes the neighbor list of node $u$, $AIE_{u,i}$ represents the value of Adjacency Information Entropy for the $i$-th neighbor of node $u$, and $D_u$ represents the degree of the node $u$.

After obtaining the node weight list $\{W_{u,v}\}_{v=1}^{Ner_u}$, the weighted neighborhood aggregation rule for node $u$ is as follows:

$$\hat{\mathbf{X}}_u^i = \sum_{v \in Ner_u} W_{u,v} \cdot \hat{\mathbf{X}}_v^{i-1} + W_{u,u} \cdot \hat{\mathbf{X}}_u^{i-1} \quad (11)$$

where $\hat{\mathbf{X}}_u^{i-1}$ denotes the perturbed data uploaded by node $u$, while $\hat{\mathbf{X}}_u^i$ represents the representation of node $u$ in the $i$-th aggregation layer.

In addition to protecting the node feature privacy, we also introduce perturbations to the node labels to protect its privacy. Given the graphs in the real world, such as those found in social networks, are homophilic [30], which means that nodes with similar structures tend to possess similar labels [31]. Based on this characteristic, we can estimate the label of node $u$ by assessing the frequency of neighbor node labels in its local neighborhood. Consequently, we use random response [32] to perturb the node labels, which introduces class-independent and symmetric noise to the labels. Specifically, this method flips labels according to the following distribution:

$$p\left(\hat{\mathbf{Y}} \mid \mathbf{Y}\right) = \begin{cases} \frac{e^{\epsilon_l}}{e^{\epsilon_l} + C - 1}, & if \ \hat{\mathbf{Y}} = \mathbf{Y} \\ \frac{1}{e^{\epsilon_l} + C - 1}, & otherwise \end{cases} \quad (12)$$

Where $\hat{\mathbf{Y}}$ and $\mathbf{Y}$ denote the perturbe and the clean labels, respectively, $\epsilon_l$ is the privacy budget, and C denotes the number of classes.

## 4 Security Analysis of TDP-GNN

In this section, we theoretically demonstrate that TDP- GNN can achieve $\epsilon$-differential privacy while offering personalized privacy protection for each node. We first present two compositional properties of differential privacy and then provide the security analysis of TDP-GNN's privacy guarantees.

**THEOREM 1.** *(Parallel Composition [33]). Let $Q_1, Q_2, \ldots, Q_\eta$ be the disjoint subsets of dataset $Q$ satisfying $Q = \cup_{i=1}^{\eta} Q_i$ and $Q_i \cap Q_j = \emptyset(\forall i \neq j)$. Let $\mathcal{B}_1, \mathcal{B}_2, \ldots, \mathcal{B}_\eta$ be a set of mechanisms where $\mathcal{B}_i(Q_i) = f(Q_i) + Laplace(\triangle(f)/\epsilon)$ provides $\epsilon_i$-differential privacy. Let $\mathcal{B}(Q) = \cup_{i=1}^{\eta} \mathcal{B}_i(Q_i)$ using independent randomness for each $\mathcal{B}_i$ and $f(Q) = \cup_{i=1}^{\eta} f(Q_i)$. Then, $\mathcal{B}(Q)$ satisfies $\max\{\epsilon_1, \epsilon_2, \ldots, \epsilon_\eta\}$-differential privacy.*

**THEOREM 2.** *(Sequential Composition [33]). Let $\mathcal{B}_1, \mathcal{B}_2, \ldots, \mathcal{B}_\tau$ be a set of mechanisms where $\mathcal{B}_i, i \in \{1, 2, \ldots, \tau\}$ provides $\epsilon_i$-differential privacy. Let $\mathcal{B}$ be another mechanism that sequentially executes $\mathcal{B}_1, \mathcal{B}_2, \ldots, \mathcal{B}_\tau$ using independent randomness for each $\mathcal{B}_i$. Then, $\mathcal{B}$ satisfies $\sum_i \epsilon_i$-differential privacy.*

**THEOREM 3.** *TDP-GNN can offer personalized privacy guarantee to each node.*

PROOF. Let $Q_1, Q_2, \ldots, Q_\eta$ be a set of mechanisms where $Q_n (X_n) = f(X_n) + Laplace(\triangle(f)/\epsilon_n)$ provides $\varepsilon_n$-differential privacy. According to Theorem 2, since $Q$ executes $Q_j$, $Q_j (X_j)$ provides $\varepsilon_j$-differential privacy. In TDP-GNN, for each node $j$, the total perturbed data submitted to the server is $\hat{X}_j = Q_j (X_j)$, where $Q_j$ is a perturbation function that satisfies $\epsilon_j$-differential privacy. Therefore, TDP-GNN can offer a personalized privacy guarantee to each node. □

**THEOREM 4.** *TDP-GNN satisfies $\epsilon$-differential privacy in the personalized data perturbation computation.*

PROOF. Let $Q_1, Q_2, \ldots, Q_\eta$ be a set of mechanisms where $Q_n (X_n) = f(X_n) + Laplace(\triangle(f)/\varepsilon_n)$ provides $\varepsilon_n$-differential privacy. Since $X_1, X_2, \ldots, X_\eta$ are the disjoint subsets of dataset $X$ satisfying $X = \cup_{n=1}^{\eta} X_n$ and $X_i \cap X_j = \emptyset(\forall i, j \in \{1, 2, \ldots, \eta\}$ and $i \neq j), Q(X) = \{Q_1 (X_1), \ldots, Q_\eta (X_\eta)\}$. According to Theorem 1, we can get that $Q(X)$ satisfies $\max \{\varepsilon_1, \varepsilon_2, \ldots, \varepsilon_\eta\}$-differential privacy. Since the privacy budget assigned to nodes ranges from $\epsilon_b$ to $\epsilon_e$, we have

$\max \{\varepsilon_1, \varepsilon_2, \ldots, \varepsilon_\eta\} = \varepsilon_e = \epsilon$. Thus, $Q$ satisfies $\epsilon_e$-differential privacy, namely, TDP-GNN satisfies $\epsilon$-differential privacy in the personalized data perturbation computation, where $\epsilon = \epsilon_e$.

□

**Theorem 5.** *TDP-GNN satisfies $(\epsilon_e + \epsilon_l)$-differential privacy.*

Proof. According to Theorem 4, the personalized data perturbation mechanism satisfies $\epsilon_e$-differential privacy, and the random response mechanism applied to node labels satisfies $\epsilon_l$-differential privacy, as described by Equation 12. In the threat model of this paper, the cloud server can directly access the graph's topology, primarily aiming to protect the privacy of both node features and labels. To achieve this goal, TDP-GNN employs personalized data perturbation to process node features privately. Subsequently, the weighted neighborhood aggregation mechanism does not disclose node features. It only performs post-processing on the perturbed node feature data and does not directly access private node features and labels. Additionally, this paper guarantees differential privacy during the training phase by perturbing node labels. Given that TDP-GNN applies personalized data perturbation and random response to each node only once, and leveraging both the basic composition theorem and the robustness of post-processing algorithms [25] for ensuring differential privacy, TDP-GNN satisfies $(\epsilon_e + \epsilon_l)$-differential privacy. □

## 5 EXPERIMENT

### 5.1 Experimental Initialization

***Datasets.*** To evaluate the performance of TDP-GNN, we trained GNN models on four classic real-world graph datasets, including three citation networks (Cora[1], Citeseer[2], and Pubmed[3]) and one social network (Facebook[4]). The specific statistics of the four datasets are shown in Table 1.

***Baselines.*** Our topology-aware differential privacy (TDP) method can be seamlessly integrated into any GNN architecture in a plug-and-play manner. We trained the GNN models using the widely adopted two-layer GCN [34] and GraphSAGE [35] architecture, respectively. To verify the effectiveness of the TDP method in the GCN architecture, we compared the model performance of Plaintext-GCN, UDP-GCN, PDP-GCN, and our TDP-GCN. The Plaintext-GCN denotes that the GNN model is trained on plaintext graph data. UDP-GCN, PDP-GCN, and TDP-GCN are DP-based GNN models. UDP-GCN and PDP-GCN are two models trained using variant DP mechanisms, where UDP-GCN is based on a uniform privacy budget allocation mechanism [18], while PDP-GCN is based on our proposed personalized privacy budget allocation mechanism. Compared with PDP-GCN, the TDP-GCN adopts the additional weighted neighborhood aggregation mechanism after the personalized data perturbation. Similarly, the comparison methods to verify the effectiveness of TDP methods in GraphSAGE architecture are denoted as Plaintext-SAGE, UDP-SAGE, PDP-SAGE, and our TDP-SAGE. The uniform privacy budget allocation mechanism employed in UDP-SAGE is based on the method proposed in [16].

---

[1]https://linqs-data.soe.ucsc.edu/public/lbc/cora.tgz
[2]https://linqs-data.soe.ucsc.edu/public/lbc/citeseer.tgz
[3]https://linqs-data.soe.ucsc.edu/public/Pubmed-Diabetes.tgz
[4]https://snap.stanford.edu/data/facebook_large.zip

**Table 1: Dataset Statistics**

| Dataset | Nodes | Edges | Classes | Features |
|---|---|---|---|---|
| Cora | 2,708 | 5,429 | 7 | 1,433 |
| Citeseer | 3,327 | 4,732 | 6 | 3,703 |
| Pubmed | 19,717 | 44,338 | 3 | 500 |
| Facebook | 22,470 | 171,002 | 4 | 4,714 |

***Settings.*** We randomly split the nodes in the dataset into training, validation, and test sets in proportions of 60%, 20%, and 20%, respectively. All models were trained using the Adam optimizer and the training epochs are fixed at 500. All experiments were implemented using PyTorch Geometric Library. The specific experimental environment utilized a server equipped with two Nvidia Tesla P40 GPUs. For evaluation metrics, we use the prediction accuracy (also known as node classification accuracy) on the test set to assess the generalization ability of the trained GNN model. To avoid randomness, each experiment is conducted 5 times. We measure the model's performance by taking the average value with a 95% confidence interval. Additionally, we set the number of privacy sensitivity levels $M = 5$, categorized as very high, high, medium, low, and very low. Furthermore, We report TDP-GNN's accuracy under two different privacy budget intervals.

### 5.2 Experimental Results and Analysis

*5.2.1* ***Effect of Topology-aware Personalized Privacy Budget Allocation Mechanism***. To evaluate the effect of the privacy budget allocation mechanism, we first conduct experiments to train the GNN models with and without the topology-aware personalized privacy budget allocation mechanism over four real-world datasets. Table 2 reports the node classification accuracy under different privacy budget intervals. It can be observed that DP-based GNN models perform worse on the node classification task compared with GNN models trained on plaintext data while providing privacy protection for sensitive information. Additionally, the difference in the performance between the UDP-GCN and PDP-GCN methods, as well as that between the UDP-SAGE and PDP-SAGE methods, illustrates that the personalized privacy budget allocation mechanism achieves higher accuracy than uniformly distributing the total privacy budget. This is because, nodes with lower node importance are more susceptible to data utility degradation under the same privacy budget setting, thereby degrading the model's accuracy. Moreover, we investigated the influence of different values of privacy budget intervals on the model performance. The results indicate that the GNN models trained with a higher privacy budget interval consistently outperform those trained with a lower privacy budget interval. This is because higher privacy budgets lead to smaller injected DP noise.

*5.2.2* ***Effect of Weighted Neighborhood Aggregation Mechanism***. To evaluate the effect of the weighted neighborhood aggregation mechanism, we conduct experiments to train the GNN model with and without the weighted neighborhood aggregation mechanism over four real-world datasets. The difference in the performance between the PDP-GCN and TDP-GCN methods, as

**Table 2: Accuracy of compared methods on the four real-world datasets. The best results are in bold.**

| GNN Architecture | Privacy Budget Interval | $Low : [\epsilon_b, \epsilon_e] = [5, 10]$ | | | | $High : [\epsilon_b, \epsilon_e] = [10, 15]$ | | | |
|---|---|---|---|---|---|---|---|---|---|
| | Dataset | Cora | Citeseer | Pubmed | Facebook | Cora | Citeseer | Pubmed | Facebook |
| GCN | Plaintext-GCN | 87.9 ± 0.37 | 84.5 ± 0.32 | 83.4 ± 0.68 | 88.4 ± 0.73 | 88.4 ± 1.22 | 84.6 ± 0.38 | 83.2 ± 1.10 | 89.2 ± 0.86 |
| | UDP-GCN | 79.3 ± 1.57 | 75.7 ± 0.61 | 69.1 ± 0.40 | 82.3 ± 0.27 | 86.7 ± 0.52 | 81.7 ± 0.32 | 71.3 ± 0.61 | 84.5 ± 0.20 |
| | PDP-GCN | 84.1 ± 1.83 | 78.8 ± 0.51 | 72.1 ± 0.52 | 84.2 ± 0.10 | 87.2 ± 0.72 | 82.3 ± 0.57 | 77.0 ± 0.48 | 85.3 ± 0.79 |
| | **TDP-GCN (Ours)** | **85.6 ± 1.27** | **80.1 ± 1.09** | **75.2 ± 0.37** | **85.1 ± 0.46** | **87.9 ± 1.08** | **83.6 ± 0.81** | **79.2 ± 0.23** | **86.4 ± 0.53** |
| GraphSAGE | Plaintext-SAGE | 97.1 ± 0.43 | 98.3 ± 0.57 | 86.1 ± 0.31 | 93.7 ± 0.36 | 97.1 ± 0.48 | 98.6 ± 0.42 | 96.0 ± 0.36 | 93.8 ± 0.19 |
| | UDP-SAGE | 84.3 ± 0.86 | 77.6 ± 0.75 | 70.72 ± 0.85 | 89.8 ± 0.31 | 91.8 ± 0.27 | 88.9 ± 0.29 | 77.8 ± 0.70 | 90.1 ± 0.41 |
| | PDP-SAGE | 88.8 ± 0.87 | 84.6 ± 0.71 | 73.5 ± 0.81 | 91.1 ± 0.31 | 93.2 ± 0.37 | 91.0 ± 0.32 | 77.6 ± 0.8 | 91.9 ± 0.28 |
| | **TDP-SAGE (Ours)** | **90.2 ± 0.74** | **86.1 ± 0.80** | **76.1 ± 0.65** | **91.5 ± 0.25** | **96.1 ± 0.16** | **93.5 ± 0.37** | **80.3 ± 0.64** | **92.4 ± 0.22** |

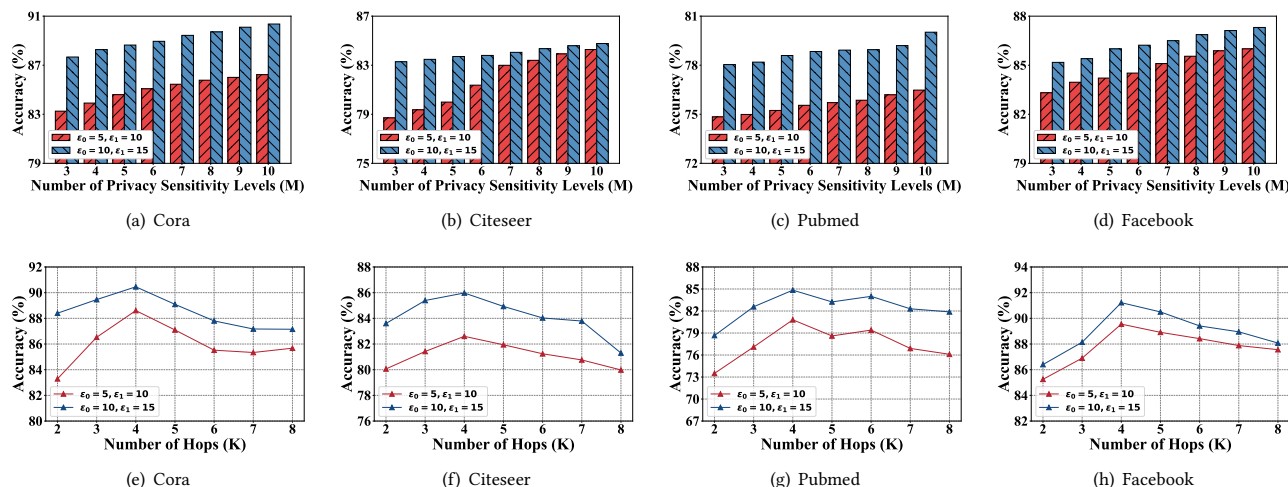

**Figure 3: GCN Architecture. The effect of the number of privacy sensitivity levels $M$ and the number of aggregation hops $K$ on the node classification accuracy of TDP-GNN.**

well as that the PDP-SAGE and TDP-SAGE methods, illustrates that the weighted neighborhood aggregation mechanism can improve the accuracy of GNN model. This is because the perturbations of different neighbor nodes are different. The proposed weighted neighborhood aggregation mechanism can effectively distinguish the differential DP noise and adaptively suppress it, resulting in a more accurate aggregated representation.

*5.2.3 **Effect of the number of privacy sensitivity levels** $M$.* We investigate how the number of privacy sensitivity levels $M$ affects the performance of TDP-GNN. We varied $M$ within $\{3, 4, \cdots, 10\}$ and reported the node classification accuracy on four datasets, as shown in Figure 3(a) - Figure 3(d) and Figure 4(a) - Figure 4(d). With the increase of $M$, the accuracy of the GNN model exhibits continuous improvement, eventually reaching a stable state. The reason is that a larger $M$ enables the TDP-GNN to provide more fine-grained personalized privacy protection for nodes in the graph, thereby reducing additional noise caused by over-protection and improving the utility of the GNN model.

*5.2.4 **Effect of the number of aggregation hops** $K$.* We analyze the effect of $K$ (referring to the layers of the GNN model in this paper) on the performance of TDP-GNN. We varied the parameter $K$ within $\{2, \cdots, 8\}$ and reported the node classification accuracy on four real-world datasets, as shown in Figure 3(e) - Figure 3(h) and Figure 4(e) - Figure 4(h). As can be seen, our method can effectively benefit from allowing multiple hops, but there is a trade-off in increasing the number of hops. With the increase of $K$, the accuracy of the TDP-GNN generally ascends to a certain threshold before stabilizing or declining. This is because as $K$ increases, the model can incorporate information from more distant nodes (i.e., all nodes within the $K$-hop neighborhood) to improve prediction accuracy. However, as the number of hops $K$ increases, the noise introduced by the aggregation operation also increases, thereby reducing the utility of the GNN model.

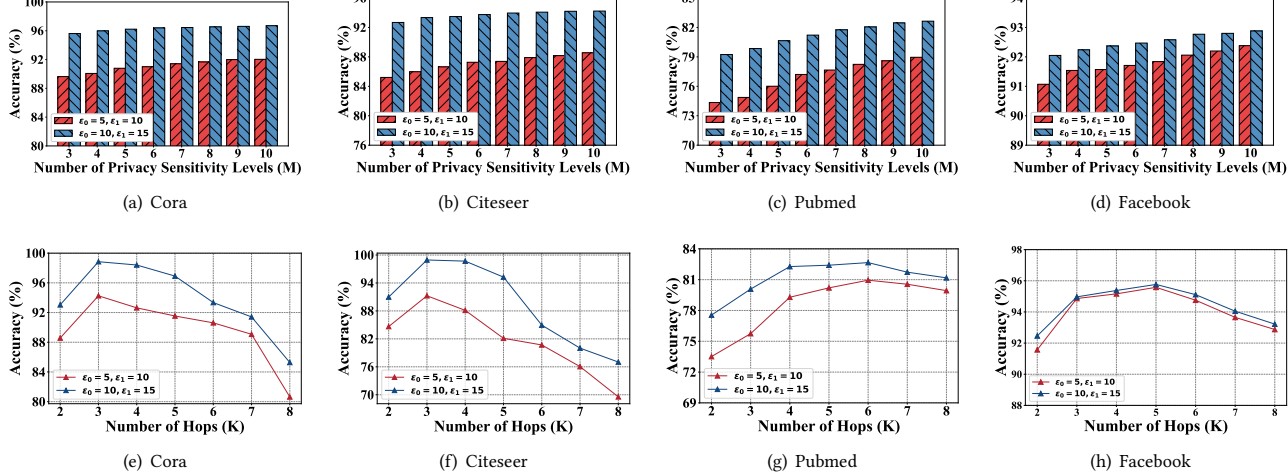

**Figure 4: GraphSAGE Architecture. The effect of the number of privacy sensitivity levels** $M$ **and the number of aggregation hops** $K$ **on the node classification accuracy of TDP-GNN.**

## 6 RELATED WORKS

Due to the advanced performance of GNNs in structured data learning, they are widely used in various graph analysis tasks [36–39]. Graph data in the real world usually contains sensitive personal information. Various privacy protection techniques have been proposed to mitigate the potential privacy leakage issue, with differential privacy standing out due to its robust mathematical underpinning. This section focuses on discussing privacy-preserving GNN learning frameworks for edge privacy and node privacy, respectively.

To protect the privacy of edges in graphs, Wu et al. [14] proposed a novel GNN architecture to achieve edge-level differential privacy. This method separates edge structures and employs only MLP to model node features and graph structural information. Kolluri et al. [15] examined adversarial link inference attacks [40] on GNNs and introduced DPGCN, a mechanism ensuring differential privacy to safeguard edge-level privacy. Zhu et al. [19] proposed Blink (Bayesian Estimation for Link-Local Privacy) to safeguard link privacy in GNNs. This approach mitigates the adverse effects of local differential privacy on GNN performance through Bayesian estimation. However, unlike the above works, this paper focuses on designing a set of mechanisms to address the privacy protection requirements of private node features across distributed nodes. In this setup, the cloud server can directly access the links between nodes, i.e., possessing global topology information, but it lacks access to the features and labels of all nodes.

To protect the privacy of nodes in graphs, Sajadmanesh et al. [12] introduced a locally private GNN learning framework within a distributed learning context. This work shares similar model assumptions with ours and primarily focuses on the privacy protection of node features and node labels. Lin et al. [13] presented a new framework named Solitude, comprising a set of mechanisms. Their work introduces multi-mechanism protection of node feature privacy by employing multi-dimensional feature perturbation and

optimizing noise through regularization terms. Chein et al. [17] proposed a novel graph learning framework: Differential Private Decoupled Graph Convolution (DPDGC), achieving a delicate balance between node attribute privacy, graph topology privacy, and GNN utility. Pei et al. [18] proposed a privacy-preserving graph neural network framework (LGA-PGNN) based on local graph enhancement. This method protects node privacy by applying local differential privacy noise to decentralized local graphs held by different data holders. Furthermore, Sajadmanesh et al. [16] proposed a differentially private GNN learning framework called GAP, which utilizes aggregation perturbation to protect node-level and edge-level privacy through the addition of noise to aggregation functions. In summary, existing works fail to consider the inherent diversity among nodes in graph learning's topological structure. They uniformly allocate the privacy budget across all nodes, thus being unable to cater to individualized privacy protection requirements for each node. This paper aims to design a topology-aware differential privacy framework to provide personalized privacy protection for each node during the GNN training process.

## 7 CONCLUSION

In this paper, we propose a personalized privacy-preserving graph neural network learning framework via topology awareness, called TDP-GNN. It can achieve accurate node representation while providing personalized privacy protection for each node. The key of TDP-GNN is a set of novel mechanisms that can allocate personalized privacy budgets based on the nodes' topological importance to satisfy their varying privacy sensitivities. In addition, a weighted neighborhood aggregation mechanism is proposed to adaptively suppress the differentiated injected DP noise, improving the utility of the GNN model. We provide theoretical analysis to demonstrate that TDP-GNN can offer personalized privacy protection for each node while satisfying $\epsilon$-differential privacy. Experimental results on four real-world graph datasets demonstrate that TDP-GNN can improve accuracy across various GNN architectures.

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
