# OpenReview forum: "Achieving Personalized Privacy-Preserving Graph Neural Network via Topology Awareness"
_ACM.org/TheWebConf/2025/Conference — WWW 2025 Poster_

### Official Review · Reviewer_APYS · 2024-11-25

**Novelty:** 4
**Technical Quality:** 4

**Review:**

# Summary
The authors propose a topology-aware differential privacy Graph neural network (TDP-GNN) learning framework via topology awareness. In TDP-GNN learning framework, a node topological importance metric and a weighted neighborhood aggregation mechanism are proposed. Theoretical analysis demonstrates that TDP-GNN can offer personalized privacy protection for each node while satisfying 𝜖-differential privacy. Experimental results on four real-world graph datasets demonstrate that TDP-GNN can improve accuracy across various GNN architectures.

# Strengths
1.	This paper is the first work that provides personalized privacy protection for nodes based on their topological characteristics in graph learning.
2.	The authors design a weighted neighborhood aggregation mechanism during the message-passing process to enhance the accuracy of GNN models trained from noising data with different levels of perturbations.
3.	This paper is reader-friendly and easy to follow.

# Weaknesses
1.	The method is not clearly introduced. In Section 3.2, the authors propose a node importance identification method based on Adjacency Information Entropy (AIE). However, this section only introduces the AIE of graph node. Hope to provide more descriptions to their node importance identification method.
2.	In Section 3.3, “suppose that the mode privacy sensitivity can be divided into M levels”. It would be appreciated if the author could provide the criteria (e.g., threshold) for determining the privacy sensitivity levels of nodes.
3.	In Section 3.5, the author allocated small privacy budget to important nodes and relatively big privacy budget to the unimportant node. However, following the power-law distribution, the majority of nodes prosses relatively small social influence. Therefore, most of the nodes would be allocated with relatively big privacy budget which may reduce the utility of GNN model. Could the authors clarify whether their intuition for privacy budget allocation could effectively address this problem?
4.	In Section 5.1, “Furthermore, We report” -> “we report”.
5.	In Section 3.5, “As described in ?? section”, the cross-reference is incorrect.

**Questions:**

My question has already been provided in the "Review-Weaknesses" section.

**Reviewer Confidence:**

3: The reviewer is confident but not certain that the evaluation is correct

**Scope:**

4: The work is relevant to the Web and to the track, and is of broad interest to the community

---

### Official Review · Reviewer_Hpos · 2024-11-26

**Novelty:** 4
**Technical Quality:** 4

**Review:**

Summary
This paper proposes a novel approach to improving the privacy and utility balance in Graph Neural Networks (GNNs) through personalized differential privacy (DP). The framework TDP-GNN addresses the problem of applying a single, uniform privacy budget to all nodes in the graph. The proposed framework allocates a personalized privacy budget based on the topological importance of each node, thus improving the utility and privacy guarantees. The method includes a topology-aware entropy measure to determine node importance and a weighted neighborhood aggregation mechanism to mitigate the impact of DP noise.

Strengths
1. Innovative Solution: The paper presents a novel solution by personalizing the privacy protection for each node based on its topological importance. This approach addresses a key limitation in existing DP-based GNNs, where a uniform privacy budget is typically applied to all nodes.
2. Utility and Privacy Balance: The proposed method shows a clear improvement in the balance between privacy and model utility, offering a more tailored approach to privacy protection.
3. Theoretical Foundations: The paper includes a solid theoretical analysis demonstrating that TDP-GNN provides 𝜖-differential privacy across all nodes, ensuring privacy guarantees.

Weaknesses
1. Limited Discussion of Privacy Trade-offs: While the paper introduces personalized privacy budgets, it could benefit from a more detailed discussion on the potential trade-offs between privacy and utility for different types of graphs or datasets.

2. Comparison with Other DP Methods: While the paper claims improvements over existing methods, a more direct comparison with other privacy-preserving graph-based methods (especially non-DP ones) could provide more context for evaluating the contribution.

**Questions:**

1. Why does Figure 2 have many "?"

2.  Although the method is theoretically sound, it would be useful to explore the scalability of the proposed approach to very large graphs. Does the personalized privacy allocation incur significant computational overhead for large-scale graphs?

3. Generalization to Non-Node-Based Privacy: The approach focuses on node-level privacy. However, some graphs might have edge-level or subgraph-level sensitive data. How might TDP-GNN be adapted to handle these cases, if at all?

**Reviewer Confidence:**

3: The reviewer is confident but not certain that the evaluation is correct

**Scope:**

4: The work is relevant to the Web and to the track, and is of broad interest to the community

---

### Official Review · Reviewer_ULsk · 2024-11-29

**Novelty:** 6
**Technical Quality:** 6

**Review:**

This paper proposes a topology-aware privacy-preserving graph neural network using individualised differential privacy budgets for the nodes in the graph incorporating their topological importance.
[+] A novel approach to provide a better privacy-utility trade-off for GNNs.
[+] Experiments conducted on four datasets show the improved accuracy of the proposed method compared to baselines
[+] A very well written paper with illustrative examples
[+] Theoretical analysis of differential privacy guarantees of the proposed TDP-GNN method.

[-] Privacy budget used in the experiments is not ideal for practical applications
[-] Baselines do not sufficiently cover the existing literature
[-] Only the accuracy metric is used for the utility evaluation. Accuracy may not be a good metric for class imbalanced datasets.
[-] Results graphs are not readable. Presentation of results could have been improved.

**Questions:**

1. The privacy budget interval used is comparatively high for real applications. Why did not the authors evaluate with a small privacy budget interval (e.g. [0.5, 2]?
2. Existing works are not compared with the TDP-GNN method. What are UDP and PDP?
3. Why did not the authors consider other utility metrics, e.g. precision, recall, f1-score, etc.? Accuracy of classification might be biased for class-imbalanced datasets

**Reviewer Confidence:**

3: The reviewer is confident but not certain that the evaluation is correct

**Scope:**

4: The work is relevant to the Web and to the track, and is of broad interest to the community

---

### Official Review · Reviewer_neFf · 2024-11-30

**Novelty:** 4
**Technical Quality:** 5

**Review:**

### Strengths
- The paper introduces a novel framework, TDP-GNN, addressing privacy-preserving concerns in Graph Neural Networks (GNNs) with topology-aware personalization.
- The structure and explanations of the methodology (node importance identification, topology-aware budget allocation, and weighted neighborhood aggregation) are clear.

### Weaknesses
- The proposed method, while innovative, may require significant computational resources, particularly for large-scale graphs.
- TDP-GCN has implemented personalized privacy budget, but also faces the problem of excessive privacy for some nodes.
-  Multiple typesetting errors, please check.

**Questions:**

- What is the purpose of $p_v$ in Eq6 ?
  - In the experimental results, it is evident that the Plain Text GCN method performs the best on all datasets, even surpassing TDP-GNN by ten percentage points. The paper only mentions: "The Plaintext-GCN indicates that the GNN model is trained on plaintext graph data."
  - In Section 3.5, there is an editing error in the phrase "As described in the ?? section"; unexpected boxes appear after the Proof of Theorem 3, Theorem 4, and Theorem 5.

**Reviewer Confidence:**

3: The reviewer is confident but not certain that the evaluation is correct

**Scope:**

4: The work is relevant to the Web and to the track, and is of broad interest to the community

---

### Official Review · Reviewer_K27A · 2024-12-02

**Novelty:** 3
**Technical Quality:** 4

**Review:**

This paper introduces a new algorithm called TDP-GNN which can achieve personalized privacy protection for each node in a graph with improved privacy-utility guarantees. In the proposed algorithm, the authors introduce a topological importance via the adjacent information entropy method for each node. Besides, the authors also implement a personalized topology-aware privacy budget to quantify the privacy sensitivity to allocate the privacy protection strength. Theoretical analysis proves that TDP-GNN achieves $\epsilon$ differential privacy, and extensive experiments prove the effectiveness of the proposed method.

Pros:
1. The paper tackles the privacy protection problem in graphs which is relevant to the Web.
2. The idea of introducing node importance based on its topological structure using entropy and using a topologically aware privacy budget to assign personalized privacy protection strength is intuitive and novel.
3. The effectiveness of the algorithm is proved through theoretical analysis and extensive experiments.

Cons:
1. While the idea is novel, the authors compare the proposed methods with not-so-competitive baselines.
2. The benchmarks used in the paper are not sufficient and are relatively small graphs.
3. The authors do not provide enough discussion on how robust the algorithm is to privacy protection and how the algorithm performs when there is an attack on privacy.

**Questions:**

1. There is a missed reference in line 497.
2. The benchmarks used in the paper are relatively small graphs, do you have more results on larger graphs or benchmarks that can better address the privacy issues?
3. Are there any experiments to show that the proposed methods can protect privacy or mitigate the risk when a privacy attack happens?
4. The experimental results shown in Table 2 indicate that nearly all the privacy protection methods fail to achieve the same performance, what is the reason?

**Reviewer Confidence:**

3: The reviewer is confident but not certain that the evaluation is correct

**Scope:**

4: The work is relevant to the Web and to the track, and is of broad interest to the community